# A Short-Term Sucrose Diet Impacts Cell Proliferation of Neural Precursors in the Adult Hypothalamus

**DOI:** 10.3390/nu14132564

**Published:** 2022-06-21

**Authors:** Antonia Recabal, Sergio López, Magdiel Salgado, Alejandra Palma, Ana M. Obregón, Roberto Elizondo-Vega, Juan C. Sáez, María Á. García-Robles

**Affiliations:** 1Departamento de Biología Celular, Facultad de Ciencias Biológicas, Universidad de Concepción, Concepción 4030000, Chile; recabalbeyer@gmail.com (A.R.); slopez@udec.cl (S.L.); magsalgado@udec.cl (M.S.); alejandrapalma@udec.cl (A.P.); relizondo@udec.cl (R.E.-V.); 2Instituto de Neurociencias, Centro Interdisciplinario de Neurociencias de Valparaíso, Universidad de Valparaíso, Valparaíso 2340000, Chile; juancarlos.saez@uv.cl; 3Escuela de Nutrición y Dietética, Facultad de Ciencias para el Cuidado de la Salud, Universidad San Sebastián, Concepción 4030000, Chile; ana.obregon@uss.cl

**Keywords:** tanycyte, neuronal precursor cell, sucrose, fructose, feeding behavior, proliferation

## Abstract

Radial glia-like cells in the hypothalamus and dorsal vagal complex are neural precursors (NPs) located near subventricular organs: median eminence and area postrema, respectively. Their strategic position can detect blood-borne nutrients, hormones, and mitogenic signals. Hypothalamic NPs increase their proliferation with a mechanism that involves hemichannel (HC) activity. NPs can originate new neurons in response to a short-term high-fat diet as a compensatory mechanism. The effects of high carbohydrate Western diets on adult neurogenesis are unknown. Although sugars are usually consumed as sucrose, more free fructose is now incorporated into food items. Here, we studied the proliferation of both types of NPs in Sprague Dawley rats exposed to a short-term high sucrose diet (HSD) and a control diet. In tanycyte cultures, we evaluated the effects of glucose and fructose and a mix of both hexoses on HC activity. In rats fed an HSD, we observed an increase in the proliferative state of both precursors. Glucose, either in the presence or absence of fructose, but not fructose alone, induced in vitro HC activity. These results should broaden the understanding of the nutrient monitoring capacity of NPs in reacting to changes in feeding behavior, specifically to high sugar western diets.

## 1. Introduction

In addition to the well-characterized adult neurogenic niche in rodents, such as the dentate gyrus of the hippocampus, the subventricular zone of the lateral ventricles, and lately, the hypothalamus [1,2,3,4], the conception of additional non-conventional neurogenic niches, such as the dorsal vagal complex (DVC) has broadened in the last decades [5]. The DVC region includes a set of nuclei located in the caudal brainstem, reaching the area postrema (AP) at the most posterior part of the fourth ventricle (4V), thus sharing an essential characteristic with the hypothalamus, i.e., its proximity to circumventricular organs (CVOs) [6]. This allows them to chemo-detect blood-borne nutrients and hormones, that may serve as a trophic cue for generating new neurons.

While the hypothalamus is the major regulatory center of energy balance and instinctive behaviors, such as feeding and reproductive behaviors, the DVC is the main integrative center of the autonomic nervous system, responsible for controlling the cardiovascular, respiratory, and gastrointestinal reflex functions [7]. Moreover, the functional role of hypothalamic neurogenesis has been extensively attributed to sustaining energy balance and body weight after dietary insults [2,8,9,10,11,12,13,14,15,16]. High fat has been one of the most studied causes of diet-triggered hypothalamic neurogenesis [2,8,9,11,17]. Although it is well known that Western diets include high sugar amounts, the possible effects of carbohydrates in adult neurogenesis are still not fully understood. The consumption of sugars is normally given in the form of glucose-fructose disaccharide naturally present in various foods such as fruits and honey, but nowadays more free fructose is incorporated into groceries [18,19]. Worldwide, the intake of high caloric fructose-sweetened snacks and beverages has risen proportionally with overweight [20].

The best-known peripheral effects of carbohydrate consumption are metabolic diseases. In particular, the metabolic complications attributable to excessive fructose consumption are largely due to increasing de novo hepatic lipogenesis [21,22]. Compared to glucose, fructose strongly promotes lipogenesis, not only serving as a substrate for fatty acid synthesis but also stimulating the transcriptional expression of de novo lipogenic enzymes and promoting the activation of lipogenic transcription factors by insulin non-related molecular pathways [23,24,25]. Indeed, subjects consuming fructose-sweetened beverages have decreased postprandial fat oxidation along with decreased resting energy expenditure, compared with subjects consuming an equivalent level of calories in glucose-sweetened beverages [26]. In addition, recent studies showed that these two hexoses differentially affect mitochondrial size, protein acetylation, and mitochondrial function, particularly fatty acid β-oxidation [27]. Thereby, due to their differing metabolism, glucose or fructose enriched diets may differentially contribute to physiological and pathophysiological processes.

Notably, these monosaccharides have also been shown to have opposite effects on the control of food intake in the brain: intracerebroventricular (i.c.v.) glucose administration decreases food intake, whereas centralized fructose administration increases feeding [28]. Indeed, in contrast with glucose, fructose metabolism in the CNS provokes a drop in the ATP/AMP ratio, increased protein kinase, AMP-activated (AMPK) activity, decreased Acetyl-CoA carboxylase (ACC) activity, and lowered malonyl–CoA in the hypothalamus [28]. However, it is unknown if both hexoses are similarly incorporated by nutrient-sensing hypothalamic cells and the metabolic and signaling consequences of their glial assimilation have yet to be identified.

Glucose transporters, GLUT2 and GLUT5, are localized in tanycytes, and both transport fructose [29]. Particularly, inhibition of Glut2 expression specifically in hypothalamic tanycytes decreased food intake in rats [30], suggesting an important role of fructose-sensing in feeding behavior. Tanycytes detect changes in glycemia and glycorrhachia, by directly contacting fenestrated capillaries of the median eminence (ME) and cerebrospinal fluid (CSF), respectively, and modulating the activity of neurons involved in the control of food intake [31,32,33,34]. Tanycytes are the hypothalamic neural precursor cells that proliferate and differentiate into fully functional orexigenic/anorexigenic neurons, which can be activated upon detection of peripheral metabolic changes [2,3,4,8,35,36]. The molecular mechanisms that induce tanycytes to proliferate after metabolic insults/signals need to be elucidated. A possible explanation lies in the purinergic signaling triggered after the activity of hemichannels (HC) [37]; a similar pathway that can be evoked by high glucose exposure [38].

Morphologically similar to hypothalamic tanycytes, radial glia-type cells named vagliocytes have been identified in the AP [39]. They share neural precursor typical markers [40], form a barrier at the boundary of the AP and the nucleus tractus solitarius (NTS) [41], and are able to form neurospheres [42]. Furthermore, in the NTS of adult mice, newborn neurons have been detected by BrdU incorporation and doublecortin or polysialylated neural cell adhesion molecule (PSA-NCAM) colocalization [40,43].

The impact of sugary diets on the generation of new neurons has been evaluated. It has been shown that fructose, but not glucose consumption reduces hippocampal neurogenesis [44]. Behavioral studies showed that the fructose-induced decline of neurogenesis leads to an impairment in cognitive functions [45], highlighting the impact of the fructose intake on neurogenesis-driven behavioral effects. However, the consequences of a high sugar diet on brain regions neurogenesis in direct contact with blood or CSF, such as the hypothalamus and DVC, are mainly unknown.

Here, we investigated the proliferation of rat tanycytes and vagliocytes exposed to a short-term high-sucrose diet (HSD; glucose plus fructose), compared to a branched low-absorption polysaccharide starch diet used as a control. Our results showed that these diets applied over 8 days did not affect body weight. Moreover, only HSD induced a high glycemic condition. We detected an enhanced proliferation of β-tanycytes and AP Nestin+ cells in rats dietary exposed to high sucrose. In tanycyte cultures, we were able to separate the effects of the monosaccharides on HC activity and detected that glucose, either in the presence or absence of fructose, but not fructose alone induces HC activity. These results will broaden the understanding of the nutrient monitoring of NPs in reacting to changes in the feeding behavior, specifically to high-sugar Western diets.

## 2. Materials and Methods

### 2.1. Ethics Statement

Animal experiments were carried out in accordance with the ARRIVE guidelines (https://arriveguidelines.org, accessed on 1 July 2021). All the studies that were carried out using rats were approved and reviewed by the animal ethics committee of the research and development national agency (ANID, N°1180871), by the ethics committee of the faculty of biological sciences of the Universidad de Concepción, Chile. Sprague Dawley adult rats were housed on a 12 h light/dark cycle with free access to food and water. Data were not pre-registered; all information is given in the manuscript and original data will be made available upon request.

### 2.2. Primary Culture of Tanycytes

The primary cultures of rat tanycytes were performed using the previously described method (Orellana, 2012, García et al., 2003). Briefly, rats at postnatal day 1 were rapidly decapitated, the brains were removed, and the region around 3V was carefully dissected. Then were incubated with trypsin 0.25%—EDTA 0.2% (*w*/*v*) for 20 min at 37 °C, disaggregated by gentle trituration and transferred to a MEM culture medium (GibcoTM), supplemented with 10% fetal bovine serum (FBS), L-glutamine, penicillin, and streptomycin (Thermo Fisher Scientific, Waltham, MA, USA). The tissue samples were disaggregated by gentle trituration, and the cells were seeded in T25 culture flasks covered with poly-L-lysine 0.2 mg/mL (Sigma-Aldrich, St. Louis, MO, USA) at 1.2 × 10^5^ cells/cm^2^ density. The cells were incubated in the same flask for two weeks, with culture medium changes every two days. Later, the monolayer-grown tanycytes were washed with PBS (pH 7.4) and incubated in a disgregation buffer containing trypsin/EDTA 0.22/0.2 (*w*/*v*) for 3 min at 37 °C. Disaggregated cells were reseeded in 6-, 12- and 24-well plates, previously covered with 0.01% (*v*/*v*) poly-L-lysine, at a density of 50,000 cells/cm^2^. Twenty-four h before experiments, cultures were maintained in an FBS-free medium to prevent inhibition of ATP release and activation of purinergic signaling.

### 2.3. BrdU Injection in Rats Subjected to High Sucrose Diet×

The rats were fed with chow diet prior to study initiation. 6–8 male rats were exposed for 8 days to a diet rich in sucrose (Cat# 58R1 AIN-76A w/11% Fat Energy/Sucrose/Blue), and to a control diet (Cat#58R0 AIN-76A w/11% Fat Energy/Corn Starch/Yellow), both obtained from Test Diet. Diets were stored at 6 ± 4 °C. The diet rich in sucrose, contains 74.3% of its energy (kcal/g) made up of carbohydrates, sucrose being 61%. The control diet contains 73.7% of its energy in carbohydrates, but the percentage of sucrose is completely replaced by cornstarch. One day after starting the change in diet, each rat was injected intraperitoneally (ip) with 200 μL of 50 mg/kg BrdU once a day for 7 consecutive days [46]. Body weight and the amount of food eaten were monitored daily. Before and after treatment, glycaemia was measured after 4 h of fasting with an Accu-Check device, puncturing the tip of the tail. One day after the last injection of BrdU, the rats were anesthetized with a mixture of ketamine: xylazine (150:15 mg/kg, respectively) and vascularly perfused with 200 mL of ice-cold PBS (17 mM Na_2_HPO_4_, 83 mM NaH_2_PO_4_ 2H_2_O, 15 mM NaCl, pH 7.4) followed by 200 mL of 4% paraformaldehyde dissolved in PBS, pH 7.4. The brains were removed, and the hypothalamus dissected for subsequent immunohistochemistry.

### 2.4. Immunohistochemistry

Hypothalamus frontal sections of 40 µm were obtained from the rat brains previously fixed by vascular perfusion and post immersion fixation in 4% (*w*/*v*) paraformaldehyde for 24 h. Slices were obtained using a vibratome, and processing by free-floating immunohistochemistry. Slices were washed with Tris phosphate buffer (84 mM Na_2_HPO_4_, 35 mM KH_2_PO_4_, 120 mM NaCl, 10 mM Tris, pH 7.8). BrdU detection was carried out using an additional treatment for allowing DNA denaturation. Samples were incubated in 1 M HCl at 45 °C for 30 min; after that were neutralized by washing the slides three times with Tris phosphate buffer. The following primary and secondary antibodies were used: mouse anti-nestin antibody (1:1500, Abcam Cat# AB6320), chicken anti-vimentin antibody (1:400, Millipore Cat# MAB3400), sheep anti-BrdU antibody (1:1000, Abcam, Cat# AB1893), goat anti-sheep Cy3, donkey anti-chicken Cy2, donkey anti-mouse Alexa 488. All secondary antibodies used were obtained from Jackson Immuno Research and used at a 1:200 dilution. Nuclei was labeled with TOPRO-3 (1:1000, Invitrogen).

### 2.5. BrdU Positive Cell Count

Two researchers participated in the counting of BrdU-positive cells for the different conditions, considering cells with positive TOPRO-3 nuclei. The final value was calculated as the average of both counts if they had an error of less than 15%. For these analyses, the overlap of all the focal planes was used in the maximum intensity projection mode of the ImageJ program (nih.gov). At least 12 slices per animal and 6 animals for each condition were used. For this analysis, images taken at 20× objective were used, and the total tissue volume was considered, excluding the ventricular area.

### 2.6. Glycemia Measurement

Blood glucose levels were measured in blood drops from the lateral tail vein using an Accu-Chek Go glucometer (Roche, Basel, Switzerland). Basal blood glucose concentration was assessed after 6 h of fasting (starting at 9 a.m.) before starting and ending treatment with the diets. Acute changes in the glycemic induced by the diets were evaluated after 10 h of fasting at 30, 60, 90, and 120 min after starting refeeding.

### 2.7. Ethidium Uptake and Fluorescence Imaging

For time-lapse dye uptake analysis, tanycytes cultured on poly-D-lysine cover glass coverslips were washed twice in phosphate-buffered saline (PBS) solution, pH 7.4, and grown to ~80% confluency for 24 h. For fluorescence detection, cells were washed twice with PBS, pH 7.4, before applying the registration solution (in mM: 154 NaCl, 5.4 KCl, 2.3 CaCl_2_, and 5 HEPES, pH 7.4), containing 5 μM ethidium bromide (Etd^+^). Cells on coverslips were mounted under a live cell microscope (Nikon, Eclipse Ti-FL model) and were recorded every 30 s with a 40× objective. In all experiments, basal fluorescence was recorded for 5 min (in 2 mM glucose registration solution). Then, the effect on Etd^+^ uptake of sugars and combinations of sugars with and without La^3+^ was tested for an additional 10 min. At least 10 cell nuclei per coverslip were defined as regions of interest (ROIs), and the average change in their fluorescence intensity was measured over time.

### 2.8. Statistical Analysis

We analyzed the data obtained from animals using the t-Student test, and Statistical differences were considered significant when *p* < 0.05. One-way ANOVA with Bonferroni’s multiple comparison test was used by in vitro determination. Statistical significance was defined as * *p* < 0.01. For all analyses, we used the Graphpad Prism 5.0 software (GraphPad Software Inc., San Diego, CA, USA).

## 3. Results

### 3.1. Short-Term Intake of a HSD Does Not Induce a Pathologic Phenotype

Among obesity-induced diets, the Western diet made up of a high fat and sugar content is the most studied. The literature exhibits a plethora of findings related to the effects of fructose over glucose (mimicking the content of the beverage), i.e., to activate the rewarding system, or prevent the fatty acid oxidation (FAO) in the liver. In the present study, we investigated the effects of a sugary caloric diet on NP proliferation in brain areas related to nutrient sensing, namely the hypothalamus and DVC. For that, two animal groups were fed a HSD, where 61% of the total carbohydrates were made up of sucrose, and a control diet (CD), which contained the same proportion of carbohydrates (around 74%) but replacing the total sucrose percentage (61%) per cornstarch. Cornstarch is a ramified oligosaccharide, whose digestion involves the breaking of high molecular mass molecule complexes into simpler ones; therefore, the CD is more slowly absorbed compared to the HSD. Short-term intake of a HSD might imply both the remodeling of the hypothalamic and DVC cytoarchitecture, as well as known metabolic complications that are not usually observed after the intake of an equivalent amount of glucose, i.e., insulin resistance, hyperglycemia, and increased insulin levels [27]. Thereby, we initially compared overall changes in glycemia, food intake, and body weight between the HSD and CD groups over 8 days (Figure 1A). Glycemia measured after 4 h of fasting, did not vary significantly between groups, ranging from 6.2 ± 0.4 mM and 6.9 ± 0.5 mM (mean ± sem) at the regime onset, to 7.5 ± 0.4 mM and 7.1 ± 0.4 mM after treatment completion, for CD and HSD groups, respectively (Figure 1B). Likewise, no significant differences were observed on body weight gain (mean ± sem; 54.9 ± 1.8 and 47.7 ± 3.5 g for CD and HSD, respectively at the end of treatment) (Figure 1C) and food intake (Figure 1D), measured regularly throughout the treatment. HSD over 8 days was not sufficient to induce obesity or diabetes in Sprague Dawley rats. Subsequently, we tested the effect of the HSD to produce a short-term rise in blood glucose compared to the CD. Animals were fasted for 10 h and then, the glycemia was analyzed before feeding and at 30, 60, 90, and 120 min after their first meal (Figure 2A). As expected, the HSD group exhibited greater glycemia values than the CD at 60 (mean ± sem; 188.2 ± 5.2 mM vs. 159.2 ± 7.7 mM; *p* < 0.05), 90 (199.0 ± 5.9 mM vs. 175.8 ± 7.7 mM; *p* < 0.05) and 120 min (205.6 ± 7.8 mM vs. 170.8 ± 4.1 mM; *p* < 0.01) (Figure 2B).

### 3.2. ME and Arcuate Nucleus (ARC) Cell Proliferation Remain Unresponsive to HSD

Subsequently, we evaluated the constitutive cell renewal in the hypothalamus and explored the effects of HSD on it. Male rats weighing 180 to 230 g were injected with 50 mg/kg ip of BrdU daily for 7 days at the end of which the brains were collected, and the hypothalamus was dissected. Forty µm thick sections from the medial basal region were selected (Bregma coordinates −1.47 to −1.79, corresponding to the periventricular ARC region). By immunohistochemical approaches, the presence of BrdU was assessed in vimentin-positive cells of the ventricular border (Figure 3A1–B2) as well as in the lateral hypothalamic section and ME of individuals subjected to CD and HSD (Figure 3A1,A2,B1,B2, respectively). BrdU-positive cells were also observed in the pars tuberalis (not shown). Since the ME contains fenestrated capillaries, it is the hypothalamic region most exposed to variations in the concentration of circulating metabolites, which can even cause structural changes in its vascularization state [47]. Therefore, the proliferative state of resident ME cells was analyzed in the HSD and CD groups. No significant changes were detected in the total BrdU-positive cells, with (mean ± sem) 58.7 ± 7.3 vs. 60.6 ± 6.3 BrdU+ cells and 0.8 ± 0.1 vs. 1.0 ± 0.2 BrdU+ cells per tissue volume for HSD and CD, respectively (Figure 3C). The amount of BrdU in cells of the ARC was similar between the HSD and CD groups, reaching up to 42.5 ± 8.0 vs. 34.3 ± 6.1 BrdU+ cells and 1.0 ± 0.2 vs. 1.0 ± 0.3 BrdU+ cells per µm^3^, respectively. Furthermore, changes in the cell division of the gross-tanycyte population in response to dietary sugar content was evaluated. For the count of Vimentin+/BrdU+ tanycytes were considered to be located up to 30 µm in the proximity of the ventricular wall. No significant changes in the proliferative state of the gross-tanycyte population were observed, with 5.8 ± 0.5 vs. 6.3 ± 1.1 BrdU+ cells for the HSD and CD group, respectively (Figure 3E).

### 3.3. HSD Differentially Boosts β-Tanycyte Proliferation

We detailed the proliferative response of each tanycyte subpopulation according to Robins et al. and Rodríguez et al. [3,48]. We first analyzed α1- and α2-tanycytes that form the dorsal and ventral portion of the 3V lateral wall, respectively. Again, no significant changes were observed in the BrdU incorporation by α1/α2-tanycytes exposed to HSD compared to CD (3.3 ± 0.5 vs. 4.8 ± 1.0 cells/30 μm, respectively) (Figure 3F). β1-tanycytes line the most basal lateral wall and a small portion of the 3V floor, whereas β2-tanycytes completely cover the 3V floor. The prolongation of the former reaches the ME, contacts the fenestrated capillaries, and forms a physical barrier that allows the permeation of systemic molecules to the hypothalamic parenchyma [41]. The β1-tanycytes were much slower dividing than the rest of the tanycyte-groups (data not shown), for which their BrdU incorporation was excluded from the quantification. β2-tanycytes responded to the rise of dietary fructose by significantly increasing their cell-division rate, from 1.5 ± 0.2 cells/30 μm under CD to 2.6 ± 0.4 cells/30 μm after HSD (Figure 3G).

Under control conditions and for reasons mentioned above, BrdU administered peripherally was differentially incorporated by subpopulations of tanycytes, of which β2 had a lower rate of cell division compared to α1/2. However, when exposed to an HSD, the proliferative rate of β2-tanycytes reached values such as that of α1/2, attenuating the proliferative differences between both subpopulations (not shown). Thus, changes in the cell division of tanycytes are a direct consequence of the increasing amount of circulating glucose after HSD or are indirectly related to HSD-induced physiological changes.

### 3.4. Characterization of the AP along the Anteroposterior Axis

We next sought to perform a serial histological map of the DVC to recognize the poorly characterized area harboring the NPs. Seven micrometer thick coronal sections of adult mice were selected every 35 μm in the anteroposterior axis, within the bregma range of −6.95 to −7.67 mm (Figure 4), according to the mouse brain stereotaxic atlas [49]. As shown in Figure 4, this region is defined by the presence of the fourth ventricle (4V) and the central canal (CC), which similar to all ventricles, are lined by vimentin+ ependymal cells (Figure 4A). The presence of the choroid plexuses identified by their characteristic cellular structure and organization in the roof of the 4V is also of note (Figure 4A1,B3). Due to nuclear staining (TOPRO-3 in blue), it was possible to recognize cell clusters corresponding to NTS and dorsal motor nucleus of the vagus (DMV) (Figure 4A2). As more posterior sections were analyzed, a clear reduction of 4V is observed (Figure 4C1), giving rise to the CC (Figure 4C1–E3); likewise, the disappearance of the choroid plexuses and the emergence of the AP (Figure 4C3), where a significant number of vimentin+ vagliocytes reside [39,40]. According to the previous analysis, the coordinates that comprise the entire AP structure are in the anteroposterior axis, between the −7.47 and −7.67 mm bregma for the mouse brain, and −13.28 and −14.68 mm bregma for the rat brain, according to the mouse and rat stereotaxic brain atlas [49].

### 3.5. Increased Proliferative Response of AP Cells Exposed to HSD

The effect of the HSD on DVC was evaluated in the same animals used for the previously described hypothalamus analysis to reduce the experimental *n*. For this aim, the posterior midbrains (including cerebellum) were dissected and sections between −13.68 and −14.68 mm bregma were selected. The sections were processed by immunohistochemistry to identify precursor cells with cumulative proliferative activity (Nestin+/BrdU+ cells) in the parenchyma of nearby neuronal nuclei (Figure 5A–F2). Cell clusters with these characteristics were found in DMV (Figure 5A,D), AP, and NTS (Figure 5B,E), as well as at the dorsal roof of the CC (Figure 5C,F) for the CD and HSD, respectively.

BrdU+ cells were quantified in at least four coronal sections of the posterior midbrain of animals subjected to each feeding condition (Figure 5; *n* = 6 per group) and the data reported was calculated over the volume analyzed (in μm^3^) and normalized to CD (mean ± sem; Figure 5G–I). The quantification initially considered BrdU+ cells in the entire DMV (Figure 5G), although no significant differences between conditions were observed (1.00 ± 0.12 and 1.02 ± 0.16 BrdU+ cells/μm^3^ for CD and HSD, respectively). Likewise, the proliferation of the vimentin+ ependymal cells lining the CC was analyzed (Figure 5H), also resulting in no significant differences between both conditions (1.00 ± 0.27 and 0.92 ± 0.11 BrdU+ cells/μm^3^ for CD and HSD, respectively). Of note, the ependymal cells forming the CC were nestin−/vimentin+, while tanycytes and vagliocytes are nestin+/vimentin+. As carried out in the hypothalamus, it was decided to classify into subgroups; on the one hand, NTS and DMV cells, and in the second group, AP cells. Identical values were found in the cell division rate of the NTS + DMV group when exposed to each treatment (not shown) (0.99 ± 0.14 for both CD and HSD). On the other hand, the proliferation rate of AP cells was 0.97 ± 0.18 and 1.36 ± 0.38 BrdU+ cells/μm^3^ for CD and HSD, respectively (Figure 6I), thus showing a significant sensitivity in their proliferation state when exposed to HSD.

### 3.6. Fructose Does Not Alter HC Activity and ATP Release

A high concentration of glucose is known to increase connexin 43 HC opening probability and ATP release in tanycyte cultures [38]. Moreover, purinergic signaling triggered by these events is thought to play a role in the self-renewal capacity of tanycytes. Since the HSD content is catabolized into glucose and fructose, we next sought to determine in vitro, if fructose increases HC activity and ATP release as glucose does. Therefore, tanycyte cultures were cultured in the absence of FBS and subjected to 2 mM glucose for 24 h, mimicking low glycemic ranges before adding 10 mM glucose, 10 mM fructose, or a mix of 5 mM each (imitating sucrose) for 15 min (Figure 6A1). Additionally, 5 mM sucrose was used as osmolarity and metabolic negative control. Extracellular ATP was measured by Luciferase/Luciferase assay, adjusted to the total protein concentration of the respective sample, and then normalized to the sucrose condition. Although no significant changes were observed between treatments, ATP values derived from the 10 mM fructose condition remained under the control threshold (0.8 ± 0.04; Figure 6A2) as compared to glucose, and glucose plus fructose conditions. Moreover, a decreased ATP release can be associated with a persistent HC closed state, which can be measured by Etd^+^ uptake recordings over time (Figure 6B1). Starting from 2 mM glucose as the basal condition, the addition of 10 mM glucose, but not 10 mM fructose, successively increased Etd^+^ uptake from min 5 and on compared to the basal value (black arrow in Figure 6B2). In line with this, 5 mM glucose in the glucose/fructose mixture was enough to trigger a response similar to that evoked by 10 mM glucose alone (black arrow in Figure 6B2). For all conditions, lanthanum ion (La^3+^) was added at minute 10 and during the last min of recording to inhibit the activity of connexin HCs (white arrow in Figure 6B1,B2). The slope of each curve was measured to obtain the Etd^+^ uptake rate (Figure 6B3). As described, 10 mM glucose and 5 mM glucose/fructose conditions exhibited a significant 1.7 ± 0.2 and 1.6 ± 0.1 -fold increase compared to basal, respectively, and both returned to basal values after the application of La^3+^ (1.0 ± 0.1 and 1.1 ± 0.2, respectively). However, no significant changes were observed after treatment with 10 mM fructose compared to basal, before nor after applying La^3+^ (1.1 ± 0.1 and 0.9 ± 0.1; Figure 6B3). To check whether fructose treated tanycytes were still responsive to glucose, a boost of 10 mM glucose was applied for 5 min after their 10 mM fructose or sucrose exposure (Figure 6C2). In fact, 10 mM fructose or sucrose between 5–10 min of recording did not promote Etd^+^ uptake compared to basal (0.8 ± 0.1 and 0.7 ± 0.2 -fold increase, respectively; Figure 6C3). Nevertheless, post hoc application of 10 mM glucose for an additional 5 min (between 10–15 min of recording) was sufficient to significantly increase the Etd^+^ uptake in these cultures (1.4 ± 0.1 and 1.8 ± 0.2 -fold increase, respectively; Figure 6C3). Therefore, fructose and sucrose do not stimulate the HC activity in tanycytes in vitro as glucose does, which can be reflected in the deficient ATP release by tanycytes exposed to fructose.

## 4. Discussion

Several studies have shown the presence of adult hypothalamic neurogenesis in murine animals and its boost after exposure to different metabolic states [50], especially in a high-fat diet (HFD) [8,12,13,51]. Since the incorporated neurons mainly acquire an anorexigenic phenotype, it has been postulated that they are part of a compensatory mechanism in response to a caloric increase [9,52]. However, it is unknown if the neurogenic event is transversal to other types of hypercaloric diets. In the present work, we showed that high sucrose intake triggers the proliferation of β2-tanycytes, which have been controversially proposed as NPs of the adult hypothalamus [13,53], and of Nestin+ cells in the AP, likely representing an NP cell population in the NTS. The proliferative response seems to be exclusive to some Nestin+ populations, since the parenchymal cells of the ARC and DMV nuclei, of the ME, and cells forming the cavity wall, i.e., α-tanycytes and CC cells did not show differences in BrdU incorporation when exposed to physiologically high sucrose content. The immutable proliferative state of ARC parenchymal cells after HFD has been observed in previous studies [51], supporting our results. The control diet used in our trials contains the same caloric proportion but is stored in the form of slowly degraded branched glucose polysaccharides. The different compositions of both diets evoked different glycemia curves over time, suggesting that the cell dividing β2-tanycytes were sensitive to plasma glucose or fructose concentration, which was likely detected due to their privileged position in contacting the ME. Fasting-feeding cycles induce persistent structural remodeling of the ME, such as extended angiogenesis to lateral hypothalamic areas and development of tight junctions in α-tanycytes during fasting, and their restoration after feeding [47]. These antecedents support the plastic conception of the ME-residing cells to face changes in the metabolic state of the individual. A clear example is the morphological plasticity of the ME external zone to facilitate the release of the gonadotropin-releasing hormone to the portal circulation during the estrous cycle, which consists of parenchymal basal lamina approaching the neuronal terminals [54]. The structural plasticity of the brain’s highly permeable zones and therefore the ability to monitor glucose levels might be reproduced in the AP since it shares cytoarchitectural characteristics with the ME: (i) both are close to neuronal nuclei involved in feeding behavior, ARC and NTS, respectively; (ii) the ventricle forming walls are constituted by tanycytes and vagliocytes, that are secured to each other through tight junction proteins, i.e., ZO-1, occludin, claudin1 and claudin5 [41]; and (iii) comprise fenestrated capillaries, which are contacted by the tanycytic and vagliocytic processes [41]. In line with this, female but not male rats fed an HFD show increased proliferation and neurogenesis specifically in the ME [14]. The same research group previously elucidated the β2-tanycytes origin of nascent neurons [13]. However, our studies cannot rule out that the cell division of β-tanycytes: (i) responds to sexual dimorphism [14] since only male rats were used; (ii) precedes the inflammatory microenvironment fostered in the pre-diabetes and pre-obesity [51,55,56,57,58]; and/or (iii) is due to a glucose-sensing mechanism dependent on the sweet taste receptor Tas1r2, and not the glucose metabolism, that may likewise evoke transmissible increases in calcium signals between tanycytes [59].

Moreover, cells residing in both the hypothalamus and DVC, are endowed with the molecular machinery essential to monitor changes in glucose levels and thus participate in energy regulation circuits; tanycytes express GLUT1, GLUT2, and the glucokinase enzyme, while in DVC, a subset of GLUT2+ neurons have been detected [60,61,62]. The function of the latter is well known; these neurons stimulate glucagon and insulin production in hypoglycemic/fasting and hyperglycemic situations, respectively [61,63]. As part of the tanycytic glucosensing mechanism, connexin 43 has been shown to play an essential role, creating a broad panglial coupled network likely to amplify the detection of the metabolic stimulus [64], acting as non-docked HC, and regulating the release of signaling molecules, such as ATP, for activating purinergic signaling [38]. Taking these milestones into account, we wondered if fructose was able to open HCs as glucose does.

In tanycytes, induction with 10 mM glucose enhanced current flow through connexin 43 HCs, without affecting connexin 43 levels on the cell surface [38]. Indeed, these effects were replicated in the present study. Glucose-induced HC opening is attributable to the extensive phosphorylation sites of the connexin 43 C-terminal portion by various kinases, which underlies the regulation of its open/close state, distribution, and degradation [65]. The 5 mM glucose and fructose mix, mimicking the already digested disaccharide, increased the Etd^+^ uptake to similar 10 mM glucose levels. In both cases, the application of La^3+^ an HC blocker was enough to block Etd^+^ uptake rate, validating the HC contribution. However, 10 mM fructose did not evoke Etd^+^ incorporation by tanycyte cultures, and although the results were not significant, it also prevented ATP release. As a proof of viability, 10 mM glucose application after 10 mM fructose indeed triggered HC activity. The observed results can be consequences of the glucose/fructose differential metabolism and activated pathways. Dietary sugars are known to increase hepatic malonyl-CoA levels, which is an intermediate of de novo lipogenesis. Since fructose induces higher rates of de novo lipogenesis than glucose [66], it is reasonable to expect that fructose leads to a more profound decrease in FAO, and hence, in ATP synthesis. Recent studies showed that an HFD plus fructose, but not glucose, increases the number of reduced size mitochondrial, reduces ATP and NADH levels (as a consequence of an FAO decline), and increases ROS levels [27]. Interestingly, fructose application to isolated mitochondria reduced FAO, implying that fructose directly inhibits mitochondrial beta-oxidation [67], independent of its effects on malonyl-CoA activity and even in an insulin-free environment [27]. In fact, fructose reduces hepatic FAO altering the transcriptional and post-translational modifications on mitochondrial proteins, i.e., inducing a hyperacetylation of ACADL and CPT-1A enzymes [27]. Physiologically, HFD/fructose, but not HFD/glucose treated mice developed hyperinsulinemia with increased basal glycemia and fasting HOMA-IR score, and impaired glucose tolerance [27]. On the other hand, hypothalamic microglial cells show an increase in proliferation and activation in response to a chronic HFD (i.e., 20 weeks), contributing to a chronic inflammatory process, which alters neuronal function [68]. In this regard, it is known that pro-inflammatory molecules, such as lipopolysaccharides and arachidonic acid, inhibit connexin gap junctions and stimulates HC activity and ATP release in some cell lines [69]. The possibility that the increased sucrose-triggered NP proliferation and the paucity of fructose HC activity witnessed here is related to an inflammatory process cannot be excluded but is unlikely due to the short-term exposure (1 week) given that more than 2 weeks was required in a separate study [70].

Future studies could consider specifying the phenotype acquired by β-tanycytes once their proliferation is induced with the HSD in order to elucidate their differentiation into glial cells (astrocytes) or neuronal cells (AgRP or POMC, among others). Our findings suggest that the hypothalamus has an adaptative mechanism to compensate for the consumption of a short-term hypercaloric diet, which is influenced by the type of sugar and its metabolization.

## Figures and Tables

**Figure 1 nutrients-14-02564-f001:**
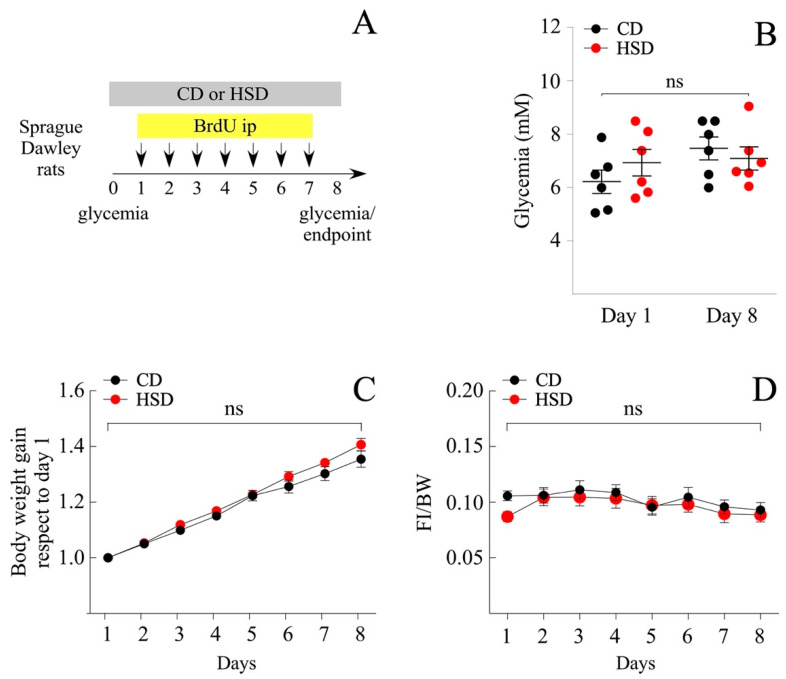
Analysis of metabolic parameters of male rats subjected to HSD. Daily ip injections of BrdU, simultaneous to the exposure of rats to a diet rich in sucrose (HSD) and to a control diet (CD), were carried out as indicated in (**A**). (**B**) Glycemic values (mM) at the beginning and end of the experiment, fasting the rats for 4 h prior to measurement. *n* = 6 animals per condition. (**C**) Body weight gain over time with respect to day 1 and (**D**) food intake over time with respect to body weight. *n* = 9 animals per condition. The data is represented as the mean ± sem; *p* < 0.01; Student’s *t*-test.

**Figure 2 nutrients-14-02564-f002:**
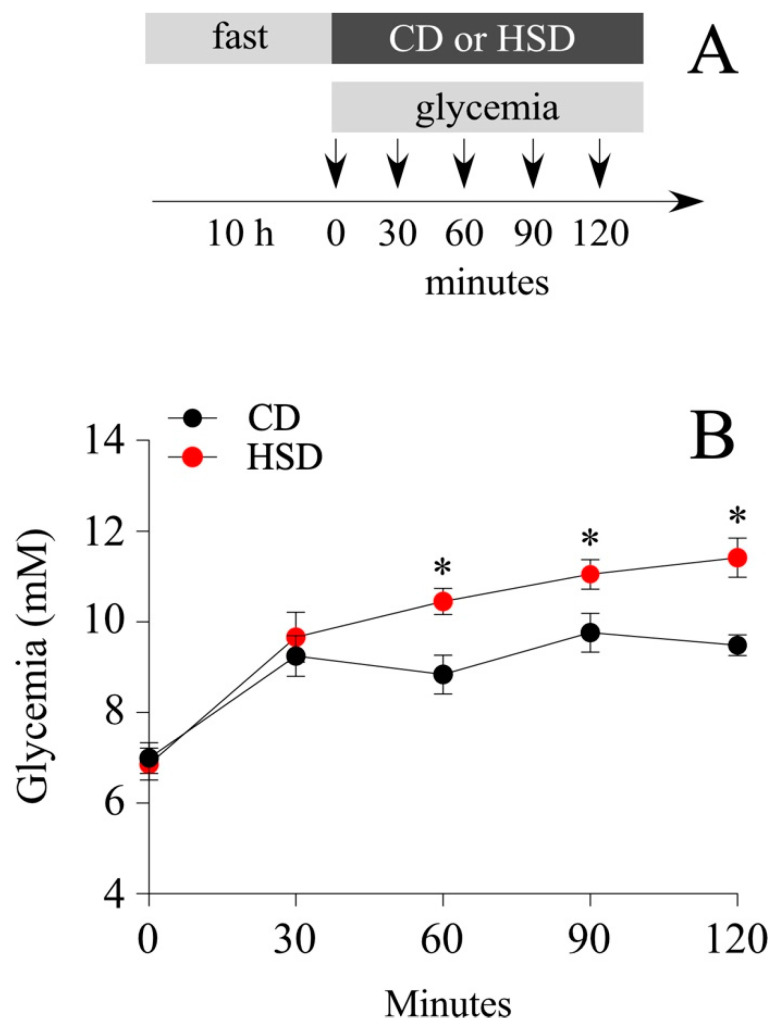
HSD-induced short-term glycemic changes. (**A**) After a 10 h fast (9 a.m. to 19 p.m.), male rats were fed ad libitum on the HSD and CD diets. Their glycemia was measured before and after feeding. (**B**) Variation in glycemia (mM) before and 30, 60, 90 and 120 min after the administration of the HSD and CD diets. *n* = 5 animals per condition; the data is represented as the mean ± sem; (*) *p* < 0.05; *t*-Student test.

**Figure 3 nutrients-14-02564-f003:**
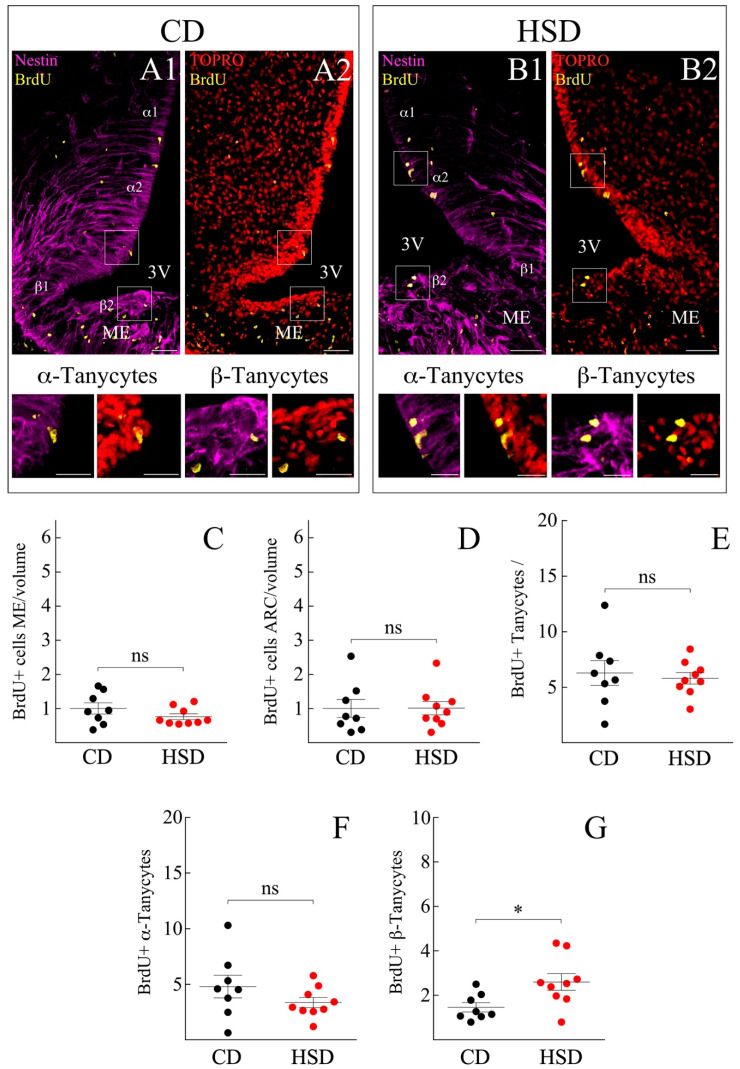
Hypothalamic cell proliferation of rats exposed to HSD and CD. (**A1**–**B2**) Immunohistochemistry of frontal hypothalamus sections of rats exposed to a control diet (CD; (**A1**,**A2**)) and a diet rich in sucrose (HSD; (**B1**,**B2**)); with antibodies that recognize nestin (purple), BrdU (yellow) and their co-localization with the nuclear marker TOPRO (red). The images in the lower panel show a magnification of the insets in (**A1**–**B2**), pointing to the proliferation of the α2 and β2 tanycytes, respectively. Scale bar (**A1**–**B2**): 50 µm. Scale bar insets: 20 μm. 3V, third ventricle. ME, Median Eminence. (**C**–**G**) Quantification of the number of dividing cells in ME (**C**) and in ARC parenchyma (**D**) normalized to the volume of tissue analyzed and relative to the control condition. (**E**–**G**) Number of proliferative tanycytes, without distinction of subpopulations (**E**), number of proliferative α tanycytes (**F**) and β2 (**G**). *n* = at least 26 slices and 8 animals for each condition. The data is represented as the mean ± sem; (*) *p* < 0.05, Student’s *t*-test.

**Figure 4 nutrients-14-02564-f004:**
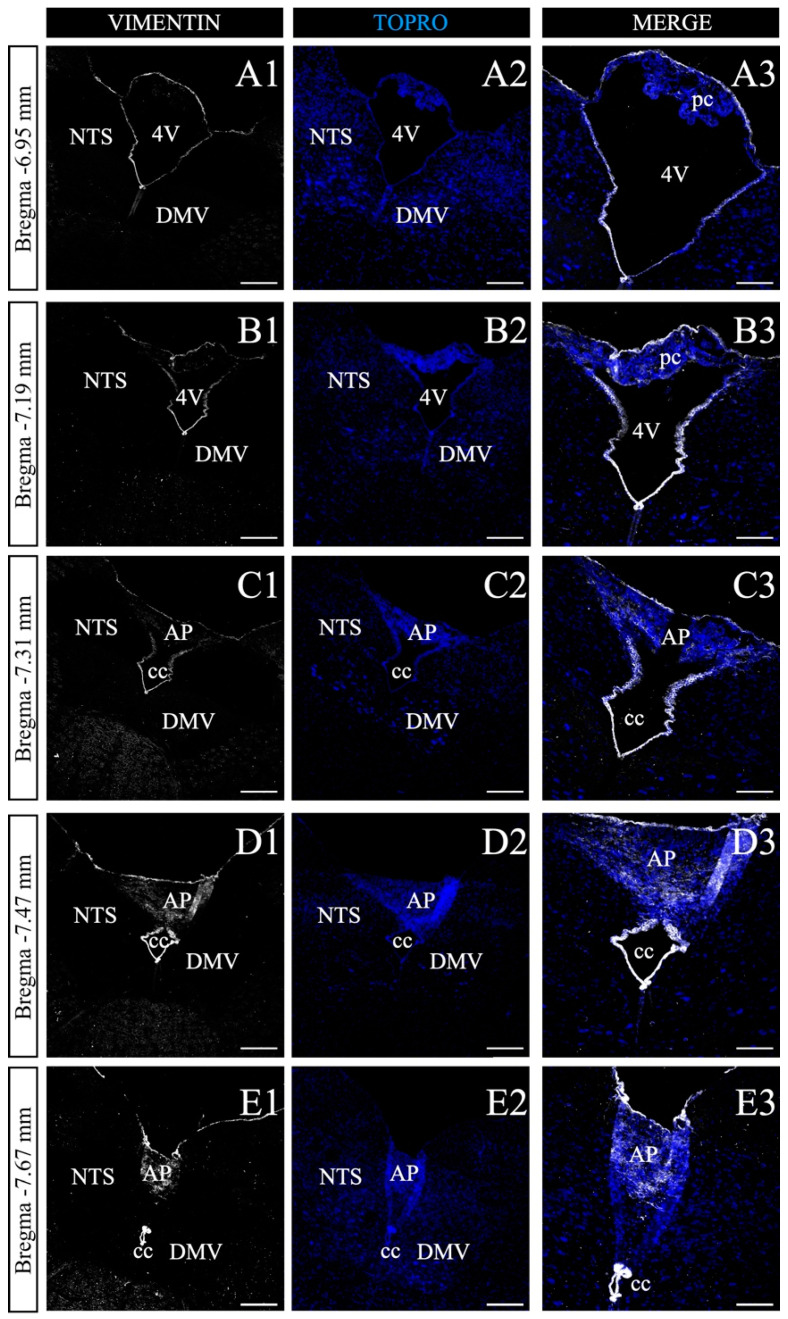
Antero-posterior screening of vimentin+ cells in the dorsal vagal complex. (**A1**–**E3**). Representative coronal sections of the mice posterior midbrain, between bregma −6.95 and −7.67 mm. Sections were processed with anti-vimentin (white), and TOPRO-3 nuclear marker (blue). Vimentin staining along the anteroposterior axis label the choroid plexuses (pc; (**A1**–**B3**)), the area postrema (AP; (**C1**–**E3**)), as well as the fourth ventricle (4V; (**A1**–**B3**)) to form the central canal (cc; (**C1**–**E3**)). Vimentin+ cells in the AP labels vagliocytes. Augmented images are shown in column 3, respectively. NST, the nucleus of the solitary tract. DMV, the dorsomedial nucleus of the vagus. Scale bar of columns 1, 2 and 3: 200 and 120 μm, respectively.

**Figure 5 nutrients-14-02564-f005:**
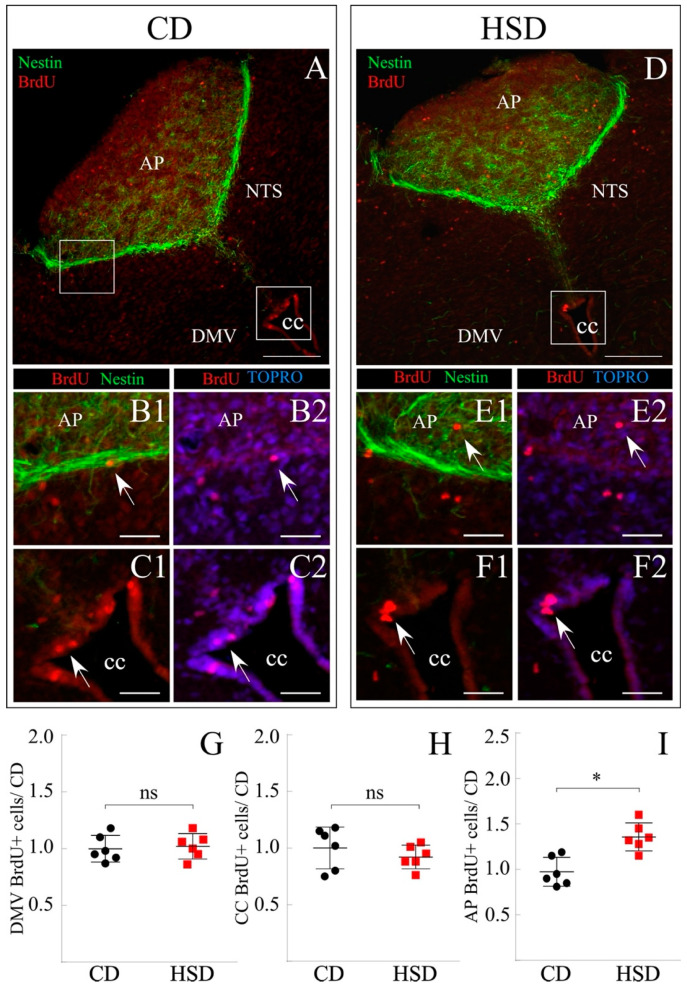
Effect of the sucrose-rich diet on cell proliferation in DVC. (**A**–**F**) Representative DVC coronal sections of rats subjected to CD (**A**–**C2**) and HSD (**D**–**F2**). (**A**–**D**) Overview of posterior midbrain immunostained for nestin and BrdU. (**B1**–**C2**,**E1**–**F2**) Augmented images of insets in (**A**,**D**), respectively, show proliferating cells in the AP and CC. AP, Area Postrema. NTS, Nucleus of the solitary tract. DMV, Dorsomotor nucleus of the vagus. CC, Central canal. Scale bar (**A**,**D**): 200 μm, (**B1**–**C2**,**E1**–**F2**): 80 μm. (**G**–**I**) Quantification of the double-positive Nestin/BrdU cell number at the DMV (**G**), CC (**H**), and AP (**I**) of rats subjected to each diet. The number of BrdU+ cells was relativized to the considered volume (excluding non-cellular areas such as the central canal) and normalized to the CD condition. The data is represented as the mean ± sem. *n* = 6. (*) *p* < 0.05, Student’s *t*-test.

**Figure 6 nutrients-14-02564-f006:**
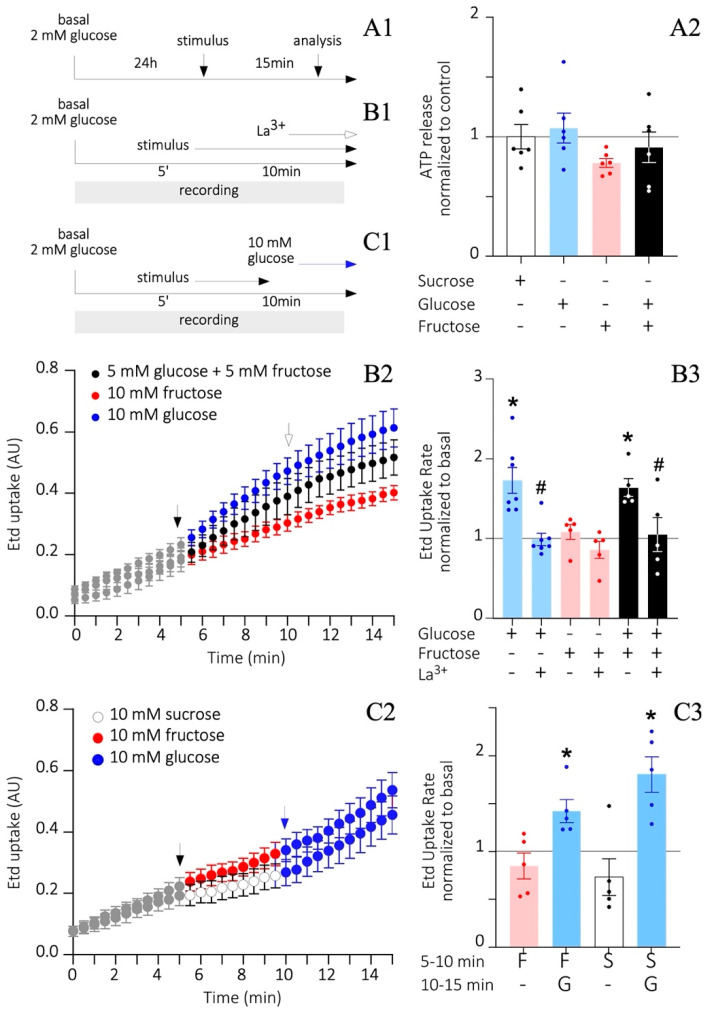
Differential sugar-induced extracellular ATP release and HCs activity in cultured tanycytes. (**A1**–**C1**) A timeline describing the procedures to measure ATP release (**A1**) and ethidium (Etd^+^) uptake (**B1**,**C1**). (**A2**) Luciferin/Luciferase assay to measure the extracellular ATP amount according to the total protein content and normalized to the sucrose condition. *n* = at least 2 independent cultures per condition. One-way ANOVA with Tukey posthoc. (**B**,**C**) Etd^+^ uptake through time (**B2**,**C2**) and Etd^+^ uptake rate (**B3**,**C3**) after exposing tanycytes to different treatments and quantified as arbitrary units of emitted nuclear fluorescence (AU) over time (min). Data in (**B2,C2**) represent the average of different experimental *n*. The black arrows in (**B2**,**C2**) indicate the moment when the stimulus was added. The white arrow in (**B2**) points to the moment when the connexin HCs inhibitor, Lanthanum ion (La^3+^), was added. The blue arrow in (**C2**) defines the time when 10 mM glucose was added. *n* = at least 10 nuclei per culture and 2 independent cultures per condition were analyzed. One-way ANOVA with Bonferroni post-hoc, (*) *p* < 0.01. *T*-test (#) *p* < 0.05).

## Data Availability

All original data will be made available upon request to the corresponding authors.

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
