# Peer review of "A Short-Term Sucrose Diet Impacts Cell Proliferation of Neural Precursors in the Adult Hypothalamus"

_nutrients, 2022, doi:10.3390/nu14132564_

Round 1

Reviewer 1 Report

In this paper Recabal and colleagues evaluated the effect of a short-term high sucrose diet (8 days) on the proliferation of rat tanycytes and vagliocytes. They also investigated the impact of glucose, fructose, or mixture of both, on hemichannel activity and ATP release in primary culture of tanycytes.

The work is very interesting and it is appreciable the clear presentation of methods and results.

I have only few concerns:

1. Why 5 mM sucrose was used as osmolarity control? (line 375)

2. The authors, in Discussion (lines 502-505), state: “The possibility that the increased sucrose-triggered NPCs proliferation and the paucity of fructose HC activity witnessed here is related to an inflammatory process cannot be excluded but is unlikely due to the short- term exposure (1 week).”

Do you have any data about hypothalamic inflammatory status after this short treatment?

3. What about expression of glucose and fructose transporter after a 1 week HSD diet?

4. Did you investigate whether dietary treatment affects AMPK activation?

It could be useful add more results about early molecular hypothalamic changes associated with this short treatment.

5. A check of English is required for eliminating some grammatical errors and/or mistakes.

Author Response

  1. Why 5 mM sucrose was used as osmolarity control? (line 375).

We thank the reviewer for pointing out our mistake, we used 10 mM sucrose as osmolarity control. This was corrected in the text

  1. The authors, in Discussion (lines 502-505), state: “The possibility that the increased sucrose-triggered NPCs proliferation and the paucity of fructose HC activity witnessed here is related to an inflammatory process cannot be excluded but is unlikely due to the short- term exposure (1 week).”

Do you have any data about hypothalamic inflammatory status after this short treatment?

Longer exposure to high-sugar diets seems necessary to affect inflammatory markers; data to date indicates that more than 2 weeks is required. This comment and reference were included (Beilharz et al 2016). Line 509

  1. What about expression of glucose and fructose transporter after a 1 week HSD diet?

Sorry but currently we do not have that data, but we are working on it. Together with transport studies, we hope to generate a new publication.

  1. Did you investigate whether dietary treatment affects AMPK activation?

We agree with the reviewer that validating the AMPK activation by fructose over glucose in tanycytes would have been ideal. However, we would like to mention that it was demonstrated in a total extract of the hypothalamus, first published in 2008 (Cha et al.)  We have included a sentence to highlight this point. Line 68

It could be useful to add more results about early molecular hypothalamic changes associated with this short treatment. We were looking if we have cDNA, but unfortunately, we have only 5 days for sending the responses to reviewers.

  1. A check of English is required for eliminating some grammatical errors and/or mistake

Done, thank you very much.

Reviewer 2 Report

The article presented by Recabal et al. is written objectively and clearly, making a good description of the state of the art and justifying the objective of the work. The results are also clearly described. The results presented add to knowledge about the impact of sucrose-rich diets on the brain, specifically on neural precursor cells. However, I have a few suggestions:

1. Line 76, “CSF” should be referred to in parentheses when “cerebrospinal fluid” is mentioned for the first time, or should also be spelled out in full in line 93 without using the abbreviation.

2. Line 82, HC should be spelled out in full, as this is the first time it is mentioned in the Introduction.

3. Lines 205 and 212, I assume the authors mean “DVC” instead of “CDV”, please correct.

4. Lines 227 and 232, I assume the authors mean “CD” instead of “ND”, please correct.

5. Line 244, ARC should also be spelled out in full when first mentioned.

6. Lines 244 and 297, I assume the authors mean “ME” instead of “EM”, please correct.

7. Line 248, please remove the “μm” from the sentence “…micrometer μm thick…” as it is in duplicate.

8. Line 429, I assume the authors mean “DMV” instead of “DNV”, please correct.

9. Line 451, please replace the “y” with “and”.

10. I suggest adding a concluding paragraph at the end of the Discussion section.  

Author Response

  1. Line 76, “CSF” should be referred to in parentheses when “cerebrospinal fluid” is mentioned for the first time, or should also be spelled out in full in line 93 without using the abbreviation.

Done, thank you very much.

  1. Line 82, HC should be spelled out in full, as this is the first time it is mentioned in the Introduction.

Done, thank you very much.

  1. Lines 205 and 212, I assume the authors mean “DVC” instead of “CDV”, please correct.

Done, thank you very much.

  1. Lines 227 and 232, I assume the authors mean “CD” instead of “ND”, please correct.

Done, thank you very much.

  1. Line 244, ARC should also be spelled out in full when first mentioned.

Done, than you very much.

  1. Lines 244 and 297, I assume the authors mean “ME” instead of “EM”, please correct.

Done, thank you very much

  1. Line 248, please remove the “μm” from the sentence “…micrometer μm thick…” as it is in duplicate.

Done, thank you very much

  1. Line 429, I assume the authors mean “DMV” instead of “DNV”, please correct.

Done, thank you very much

  1. Line 451, please replace the “y” with “and”.

Done, thank you very much.

  1. I suggest adding a concluding paragraph at the end of the Discussion section,

Done, line 511. thank you very much.